# Time trends between 2002 and 2017 in correlates of self-reported sitting time in European adults

**Judith G. M. Jelsma**[1]*, **Joanne Gale**[2], **Anne Loyen**[3], **Femke van Nassau**[1], **Adrian Bauman**[2], **Hidde P. van der Ploeg**[1]*

**1** Amsterdam UMC, Vrije Universiteit Amsterdam, Department of Public and Occupational Health, Amsterdam Public Health Research Institute, Van der Boechorststraat, Amsterdam, The Netherlands, **2** Prevention Research Collaboration, School of Public Health, University of Sydney, Sydney, Australia, **3** Centre for Nutrition, Prevention and Health Services, National Institute for Public Health and the Environment, Bilthoven, The Netherlands

* j.jelsma@amsterdamumc.nl (JJ); hp.vanderploeg@amsterdamumc.nl (HV)

## Abstract

### Background

This study explores trends in the prevalence of high sitting time and its correlates among "high sitting" and "high sitting-least active" European adults from 2002 to 2017. Both groups have merit for future public health interventions to prevent development of a range of prevalent non-communicable diseases.

### Methods

Data collected in the 2002 (15 countries), 2005 (30 countries), 2013 (28 countries) and 2017 (28 countries) Eurobarometer surveys were used, including around 15,000 respondents in 2002 and >26,000 respondents in the other years. Sitting time and moderate to vigorous intensity physical activity were measured with the validated International Physical Activity Questionnaire-short. High sitting was defined as >7.5 hours per day. Respondents in the lowest quartile of total reported days of physical activity (i.e. days walking, days in moderate activity, and days in vigorous activity) were defined as least active. Multivariate odds ratios of high sitting, and high sitting-least active were assessed by country and socio-demographic characteristics for each survey year using binary logistic regression analyses.

### Results

Trends in sitting time were relatively stable over a 15-year period, although this time trend was limited by a change in the sitting question between 2005 and 2013. Men, higher educated people, students, retired people, white collar workers, people living in urban areas, people with lower physical activity levels, and people living in the Czech Republic, Denmark or the Netherlands were consistently more likely to be in the high sitting group across all four survey years. Similarly, men, students, retired people, unemployed people, white collar

**Data Availability Statement:** All Eurobarometer datasets are stored and publicly available at the GESIS – Leibniz Institute for the Social Science (formerly GESIS-ZA, Central Archive for Empirical

Social Research, Cologne, Germany; http://www.
gesis.org/eurobarometer). Reproduction is
authorized, with the exception of commercial
purposes, provided the source is acknowledged.

**Funding:** The authors received no specific funding
for this work.

**Competing interests:** The authors have declared
that no competing interests exist.

workers, and people living in the Czech Republic or Denmark were consistently more likely
to be in the high sitting-least active group across all four surveys.

## Conclusion

This study identified population sub-groups that need special attention in public health interventions to lower total sitting time.

## Introduction

Sedentary behavior is defined as 'any waking behavior characterized by an energy expenditure
<1.5 metabolic equivalents (METs) while in a sitting or reclining position' [1], for example
while traveling by car, train or bus, at work/school, and watching television. High volumes of
sedentary behavior are associated with an increased risk of all-cause mortality [2–9], cardiovascular mortality [3–5, 8, 9], cardiovascular disease [3, 5, 8] and type 2 diabetes [3–6, 8] Even
though, physical activity attenuates the risk of high volumes of sedentary behavior, very high
levels of physical activity are needed to fully compensate the risks of sedentary behavior [9].
Importantly, people who are also physically inactive (i.e. not meeting the WHO recommendations of at least 150 minutes of moderate-to-vigorous activity per week) are especially at risk
for all-cause and cardiovascular disease mortality [9].

Interest in sedentary behavior has increased rapidly over recent years. Several (inter)
national health authorities have updated their physical activity (PA) recommendations [10–
13] to include (non-specific) statements to reduce and break up sitting time. Knowledge on
population-groups most at risk, i.e. those with high sedentary behavior, and especially those
with high levels of sedentary behavior and low levels of PA, can inform public health prevention policy.

In Europe, the European Commission collects biannual cross-national population data on a
range of social, economic and health issues through the Eurobarometer survey [14]. In 2002,
2005, 2013 and 2017 questions about sitting time (i.e. proxy for sedentary behavior) were
included. Previous studies have used the 2002, 2005 and 2013 trends to suggest that time spent
in sedentary behavior was not increasing in the European region [15]. Furthermore, earlier
studies of the 2005 and 2013 Eurobarometer data suggested that high levels of sitting time
(>7.5 hours per day (h/d)) were more prominent for men [16, 17], highly educated people [16,
17], urbanized residents [17], white collar workers [17], widows [17] and people with a low life
satisfaction [17] in a single survey.

With the addition of the 2017 Eurobarometer we first studied 15-year trends in self-
reported sitting time in European adults over four surveys. And secondly, we examined factors
that were consistent correlates of high sitting time, as well as the combination of high sitting
time and low levels of PA, over all four time points. Both groups have merit for future public
health interventions.

## Methods

### Eurobarometer

We used data from the Special Eurobarometer 472 (wave 88.4; December 2017) [18], Special
Eurobarometer 412 (wave 80.2; November-December 2013) [19], Eurobarometer wave 64.3
(November-December 2005) [20], Special Eurobarometer 183–6 (wave 58.2; October-

December 2002) [21]. The Eurobarometer covers the population aged 15+ years of European Union Member States. Participants were systematically sampled from each member state using a multistage random sampling design, with multiple sampling points based on country specific population size and density (i.e. metropolitan, urban and rural areas). Households were selected as every $N^{th}$ address by standard random route procedures based on a selected initial address. All respondents were interviewed face-to-face at home and in their native language using a standardized protocol. More information on the Eurobarometer series can be found at: http://ec.europa.eu/commfrontoffice/publicopinion/index.cfm. A total of 16,230 (15 countries), 29,193 (30 countries), 27,919 (28 countries), and 28,031 (28 countries) Europeans were interviewed in 2002, 2005, 2013, and 2017, respectively. The European Commission approved the study protocols and informed consent was obtained from all respondents. All information was anonymized prior to analysis.

## Variables

**Sitting time.**    Sitting time was measured with the validated International Physical Activity Questionnaire (IPAQ)-short [22, 23] sitting item: "*How much time do you spend sitting on a usual day*? *This may include time spent at a desk, visiting friends, studying or watching television*". In the 2013 and 2017 Eurobarometer surveys participants chose a categorical response: 1 hour or less; 1 hour to 1.5 hour; — 7.5 hours to 8.5 hours (in one-hour intervals); more than 8.5 hours, and don't know. In the 2002 and 2005 Eurobarometer surveys participants were asked for the number in hours and minutes as an open-ended question. Across all four surveys, we dichotomized the sitting item into sitting $\leq$7.5 h/d and sitting >7.5 h/d, based on two meta-analyses that showed increased risk of all-cause mortality for adults sitting for >7–8 h/d, which informed the selection of this cut-point [7, 9].

**Physical activity.**    PA was assessed by the IPAQ-short, asking respondents how many days they spent in vigorous PA, moderate PA and walking in the last seven days [24]. We created quartiles of the sum of days (possible range 0–21 days) respondents reported doing PA across the three PA items (i.e. 1[st] quartile 0–5 days (low PA); 2[nd] quartile 6–7 days; 3[rd] quartile 8–12 days; 4[th] quartile >12 days (high PA)). As the different survey years used different versions of the IPAQ, time spent in these physical activities per day impaired comparability across time. However, the reported number of days in PA items were not changed between the surveys and have shown to be a good proxy of total PA levels [25, 26].

**Social-demographic variables.**    Age was coded in 10-year categories (i.e. 18–24 years, 25–34 years, — 55–64 years, and 65 years and older), marital status was recoded as single, married/de-facto, separated/divorced and widowed. Level of education was assessed by age at completion of full-time education, recoded as 15 years and less, 16–19 years, 20 years and over, and "still studying". Current occupation was recoded into eight categories: self-employed, managers, other white collars, manual worker, house persons, unemployed, retired and students. Type of community was categorized as rural/village, small/medium sized town, and large town.

## Statistical analyses

All respondents aged <18 years were excluded. All respondents who answered 'don't know' on the sitting question were removed from the analysis. For all other variables, the response options 'refusal' and 'don't know' were coded as missing values. Data presented by country is weighted according to nation weights which weight the in-country samples according to nation specific demographics. When results are presented at the European level or by groups of nations, the data is weighted according to demographic distribution/ representation in

Europe in the given survey year. More information can be found at: https://www.gesis.org/eurobarometer-data-service/survey-series/standard-special-eb/weighting-overview/. When data are compared across years, only data from a comparable minimum set of countries was used, given that the number of participating nations varied among surveys. Countries are presented in their geographic region based on the United Nations publication M49 standard [27].

Sample characteristics and sitting prevalence were analyzed with descriptive statistics. Multivariate odds ratios (ORs) of "high-sitting" and of "high-sitting/least active" (i.e. sitting>7.5h/d and lowest PA quartile) were assessed by country and socio-demographic characteristics for each survey year using binary logistic regression analyses. To evaluate time trends 'time' was considered a factor in the model. A time by correlate interaction term was included (country or other correlate) to calculate the odds ratios at each year by correlate combination. Each country is compared to all other countries. Due to collinearity between the education and occupation variables (i.e. both include the group 'student') two multivariate models were constructed, whereby only the results from the occupation variable are presented for the second multivariate model. Furthermore, we conducted a third multivariate model replacing the country variable with a variable in which the countries were clustered in four geographical regions. In the result section we consider results 'consistent' if on all available time points they show statistically significant associations in the same direction. All analyses were conducted in SAS, version 9.4. Statistical significance is indicated for p<0.05 and p<0.001.

## Results

### Sampling

We excluded 577, 1038, 610 and 493 respondents younger than 18 years for 2002, 2005, 2013 and 2017, respectively. We further excluded 983, 1556, 702 and 755 respondents that answered 'don't know' on the sitting question for 2002, 2005, 2013 and 2017, respectively. In the final analyses 14,692; 26,645; 26,617; and 26,791 respondents were included from the survey years. Characteristics of the total population, "high-sitters" and "high-sitters/least-active" are shown by socio-demographic characteristics for all four surveys in Table 1. Across surveys, gender (i.e. ~52% women) and occupation are rather stable. Across surveys there was an increase in higher educated respondents (i.e. 21%, 25%, 29%, 32% across years). In 2002, fewer respondents were married (60% compared to ~65% in the other years) and more respondents were living in a large town (39% compared to ~26% in the other years). The mean age (SD) of the population was 46.1 (17.4), 47.7 (17.5), 50.1 (17.5) and 52.0 (17.7) in 2002, 2005, 2013 and 2017, respectively. The mean number of total days of PA in the last week in the least active PA quartile is 2.4 (1.9), 2.4 (1.9), 2.5 (1.9) and 2.3 (1.0) for 2002, 2005, 2013 and 2017, respectively. In S1 Table these data are presented by country across surveys.

### Prevalence of high sitting time

Table 1 indicates that across the years 17.5–22.5% of the population was considered "high-sitting". Over time we observed higher odds of people sitting >7.5 h/d in 2002 (OR = 1.32 (95% CI = 1.20–1.44) and in 2005 (OR = 1.28 (95%CI = 1.17–1.40) compared to 2017, but we did not observe a difference in 2013 compared to 2017 (OR = 0.94 (95%CI = 0.86–1.03). The "high-sitters/least-active" population comprise 7–10% of the population across surveys. Over time we observe higher odds of "high-sitters/least-active" in 2002 (OR = 1.27 (95%CI = 1.13–1.43) compared to 2017, but we did not observe a difference in 2005 (OR = 1.00 (95% CI = 0.88–1.13) compared to 2017. Although, in 2013 we noted a lower odds of "high-sitters/least-active" (OR = 0.83 (95%CI = 0.73–0.93) compared to 2017.

**Table 1. Total population, high sitters and high sitters-least active by socio-demographic characteristics for all four surveys.**

| Survey year | 2002 | | | 2005 | | | 2013 | | | 2017 | | |
|---|---|---|---|---|---|---|---|---|---|---|---|---|
| | % total population | % sitting >7.5 h/d | % high sit/low active* | % total population | % sitting >7.5 h/d | % high sit/low active* | % total population | % sitting >7.5 h/d | % high sit/low active* | % total population | % sitting >7.5 h/d | % high sit/low active* |
| **Total population** | | 22.5 | 10.1 | | 21.1 | 7.6 | | 17.5 | 7.2 | | 19.2 | 8.7 |
| **Gender** | | | | | | | | | | | | |
| Men | 48.2 | 25.2 | 10.8 | 48.7 | 22.7 | 7.6 | 48.3 | 18.8 | 7.0 | 48.1 | 20.8 | 8.6 |
| Women | 51.8 | 19.9 | 9.4 | 51.3 | 19.5 | 7.6 | 51.7 | 16.3 | 7.3 | 51.9 | 17.6 | 8.8 |
| **Age** | | | | | | | | | | | | |
| 18–24 years | 12.8 | 27.4 | 8.0 | 13.4 | 28.1 | 7.9 | 11.6 | 18.3 | 4.4 | 10.7 | 22.2 | 6.4 |
| 25–34 years | 20.0 | 23.5 | 9.5 | 18.5 | 22.1 | 7.2 | 16.0 | 18.7 | 6.2 | 15.1 | 19.7 | 5.9 |
| 35–44 years | 18.0 | 21.4 | 8.7 | 19.7 | 19.5 | 6.2 | 18.0 | 17.1 | 5.9 | 16.8 | 18.3 | 7.7 |
| 45–54 years | 15.8 | 22.1 | 10.8 | 16.5 | 21.0 | 7.8 | 18.2 | 18.2 | 7.1 | 17.6 | 18.3 | 7.4 |
| 55–64 years | 13.7 | 18.1 | 8.4 | 13.2 | 15.8 | 6.6 | 15.0 | 16.0 | 7.3 | 15.1 | 18.1 | 9.0 |
| 65 years and older | 19.7 | 22.6 | 13.9 | 18.8 | 20.3 | 9.9 | 21.2 | 16.9 | 10.5 | 24.7 | 19.4 | 12.7 |
| **Marital Status** | | | | | | | | | | | | |
| Single | 23.1 | 28.4 | 9.9 | 19.9 | 26.9 | 8.3 | 19.1 | 19.4 | 6.1 | 20.1 | 23.4 | 8.3 |
| Married/De-facto | 60.1 | 19.4 | 8.8 | 64.7 | 18.9 | 6.8 | 65.9 | 16.5 | 6.7 | 64.4 | 17.3 | 7.7 |
| Separated/Divorced | 8.2 | 25.0 | 13.6 | 6.3 | 22.6 | 7.7 | 7.1 | 17.9 | 7.7 | 6.9 | 17.4 | 9.2 |
| Widowed | 8.6 | 25.5 | 16.0 | 9.1 | 22.2 | 11.9 | 7.8 | 19.8 | 13.5 | 8.6 | 25.0 | 16.9 |
| **Age when finished education** | | | | | | | | | | | | |
| 15 years and less | 28.7 | 17.6 | 10.3 | 27.7 | 14.6 | 6.8 | 19.8 | 13.7 | 8.0 | 17.5 | 17.8 | 11.6 |
| 16 to 19 years | 42.6 | 18.6 | 8.8 | 40.0 | 17.7 | 6.5 | 44.6 | 13.8 | 6.1 | 43.9 | 15.8 | 8.1 |
| 20 years and over | 21.4 | 31.3 | 12.6 | 25.2 | 28.2 | 9.2 | 29.4 | 23.2 | 7.9 | 32.1 | 22.4 | 7.8 |
| Still Studying | 7.3 | 38.0 | 9.1 | 7.1 | 40.4 | 10.9 | 6.2 | 27.2 | 6.1 | 6.6 | 28.3 | 6.4 |
| **Occupation** | | | | | | | | | | | | |
| Self-employed | 8.5 | 18.9 | 7.9 | 9.7 | 17.8 | 4.9 | 7.7 | 15.5 | 6.8 | 8.0 | 15.7 | 6.6 |
| Manager | 9.8 | 36.6 | 13.2 | 10.5 | 32.8 | 11.6 | 10.3 | 28.9 | 8.1 | 11.2 | 28.9 | 8.9 |
| Other white collar | 11.4 | 40.2 | 17.0 | 11.4 | 38.9 | 13.6 | 11.8 | 36.5 | 13.0 | 12.5 | 32.1 | 12.4 |
| Manual worker | 22.5 | 10.9 | 4.7 | 20.3 | 9.1 | 2.4 | 21.7 | 7.5 | 2.6 | 21.8 | 7.7 | 3.6 |
| House person | 11.9 | 11.5 | 6.1 | 12.2 | 11.2 | 5.2 | 7.9 | 5.2 | 2.7 | 5.6 | 11.1 | 7.3 |
| Unemployed | 5.5 | 19.1 | 10.0 | 5.7 | 16.0 | 5.4 | 8.3 | 9.8 | 4.7 | 6.3 | 14.7 | 6.9 |
| Retired | 23.5 | 22.5 | 13.8 | 23.3 | 19.5 | 9.4 | 26.2 | 17.2 | 10.5 | 28.0 | 19.9 | 12.7 |
| Student | 6.9 | 37.3 | 8.6 | 6.9 | 40.4 | 10.9 | 6.1 | 27.2 | 6.1 | 6.5 | 28.3 | 6.4 |
| **Type of Community** | | | | | | | | | | | | |
| Rural/Village | 32.9 | 17.9 | 9.4 | 34.4 | 17.4 | 7.2 | 32.4 | 14.5 | 6.7 | 29.3 | 17.3 | 8.4 |
| Small/Mid-sized town | 28.0 | 21.7 | 9.9 | 39.5 | 21.0 | 7.5 | 41.4 | 17.0 | 6.6 | 44.8 | 18.7 | 8.5 |
| Large Town | 39.1 | 26.9 | 10.8 | 26.1 | 26.3 | 8.5 | 26.3 | 21.9 | 8.7 | 25.9 | 22.2 | 9.3 |
| **PA Quartile** | | | | | | | | | | | | |
| First Quartile—Low PA | 31.9 | 31.6 | na | 27.2 | 28.0 | na | 33.0 | 21.8 | na | 34.5 | 25.2 | na |
| Second Quartile | 20.8 | 23.1 | na | 21.3 | 23.6 | na | 22.5 | 18.3 | na | 23.9 | 18.9 | na |
| Third Quartile | 25.7 | 19.2 | na | 25.2 | 20.8 | na | 24.7 | 16.1 | na | 24.2 | 17.6 | na |

(*Continued*)

**Table 1.** (Continued)

| Survey year | 2002 | | | 2005 | | | 2013 | | | 2017 | | |
|---|---|---|---|---|---|---|---|---|---|---|---|---|
| | % total population | % sitting >7.5 h/d | % high sit/low active* | % total population | % sitting >7.5 h/d | % high sit/low active* | % total population | % sitting >7.5 h/d | % high sit/low active* | % total population | % sitting >7.5 h/d | % high sit/low active* |
| Fourth Quartile—High PA | 21.6 | 12.3 | na | 26.3 | 12.1 | na | 19.8 | 11.1 | na | 17.3 | 9.9 | na |

Notes: Data is weighted according to demographic distribution/representation in Europe in the given survey year.

Abbreviations: h/d: hours per day; na: not applicable; PA: physical activity

*High sit/low active = sitting >7.5 hours per day and being in the least active PA quartile.

## Correlates of "high sitting time"

The results of the multivariate analyses for the "high-sitting" group are shown in Table 2. By country, the Czech Republic, Denmark and the Netherlands are consistently high sitting countries, while people in Ireland, Lithuania, Portugal, Romania and Spain have a lower odds of being in the "high-sitting" group compared to the rest of Europe; this results in southern Europe sitting less compared to all other regions. With regard to occupation, self-employed people, managers and other white collar workers, students, and retired people have a higher likelihood of high sitting than manual workers. We observed a linear trend by PA status, with more active people being less likely to be "high-sitters" compared to the lowest PA quartile. In addition, women, and those living in a rural/village or small/mid-sized town compared to a large town have a lower likelihood of high sitting. Higher educated people were consistently more likely to sit >7.5 h/d than lower educated people. Interestingly, in 2017 the odds for higher educated people was substantially lower than in the previous three surveys. Age and marital status showed less clear correlates with high sitting time across the survey years.

## Correlates of "high sitting time and least-active"

The results of the multivariate analyses for the "high-sitters/least-active" are shown in Table 3. By country, the Czech Republic and Denmark are consistently "high-sitters/least-active" countries, while Ireland and Spain have lower odds at "high-sitters/least-active". Southern European countries report lower likelihood of "high-sitters/least-active" compared to other regions. Self-employed, managers and other white collar workers, students, retired people and unemployed people show a higher likelihood of "high-sitters/least-active" compared to manual workers. Women are less likely to demonstrate "high-sitters/least-active" compared to men. Age, marital status, type of community and education were inconsistent correlates of the "high-sitters/least-active" pattern across surveys.

## Discussion

In this study we explored the trends and correlates of high levels of self-reported sitting time and the combination of high levels of sitting time and physical inactivity in European adults assessed by the Eurobarometer surveys across 28 EU Member States over four time points between 2002 and 2017. Over the 15-year period self-reported sitting time seems rather stable, although comparison is limited by a change in survey methodology between the 2005 and 2013 surveys. The earlier observed decline in high sitting time based on 2002, 2005 and 2013

**Table 2. Multivariate odds ratio (OR) of sitting more than 7.5 hours per day, by country and socio-demographic characteristics for each survey year.**

| Category | Multivariate model OR (95%CI) of sitting >7.5 hours per day | | | |
|---|---|---|---|---|
| | 2002 | 2005 | 2013 | 2017 |
| **Country** (ref: all other countries) | | | | |
| **Northern Europe** | | | | |
| Denmark | **1.66 (1.43,1.93)** | **1.36 (1.17,1.6)** | **1.84 (1.56,2.16)** | **1.72 (1.45,2.04)** |
| Estonia | | 1.17 (0.99,1.39) | **1.20 (1.02,1.43)** | **1.21 (1.01,1.44)** |
| Finland | **1.39 (1.19,1.63)** | **1.29 (1.10,1.51)** | **1.28 (1.05,1.56)** | **0.75 (0.62,0.92)** |
| Ireland | **0.61 (0.50,0.74)** | **0.51 (0.42,0.62)** | **0.48 (0.38,0.62)** | **0.65 (0.53,0.78)** |
| Latvia | | **0.78 (0.65,0.94)** | 0.98 (0.81,1.17) | 1.04 (0.86,1.26) |
| Lithuania | | **0.78 (0.64,0.94)** | **0.73 (0.61,0.87)** | **0.74 (0.61,0.90)** |
| Sweden | 1.11 (0.95,1.31) | 1.01 (0.86,1.17) | **1.53 (1.28,1.83)** | **1.53 (1.27,1.84)** |
| UK | **0.77 (0.64,0.93)** | 0.96 (0.81,1.15) | 1.10 (0.93,1.31) | 0.89 (0.75,1.07) |
| **Western Europe** | | | | |
| Austria | 1.00 (0.84,1.20) | **0.73 (0.61,0.87)** | **1.22 (1.01,1.47)** | 0.85 (0.70,1.02) |
| Belgium | 0.98 (0.83,1.15) | 1.13 (0.96,1.32) | 0.85 (0.71,1.02) | 1.01 (0.84,1.21) |
| France | **0.59 (0.50,0.70)** | **0.59 (0.49,0.71)** | 0.94 (0.79,1.12) | **0.72 (0.60,0.87)** |
| Germany | **1.25 (1.09,1.44)** | **1.22 (1.06,1.42)** | **1.20 (1.02,1.41)** | 1.15 (0.98,1.35) |
| Luxembourg | **1.29 (1.04,1.60)** | 1.02 (0.77,1.35) | **1.39 (1.08,1.79)** | 0.93 (0.71,1.21) |
| Netherlands | **1.40 (1.16,1.67)** | **3.00 (2.58,3.48)** | **2.41 (2.05,2.83)** | **3.14 (2.69,3.67)** |
| **Eastern Europe** | | | | |
| Bulgaria | | **0.82 (0.68,0.98)** | 1.01 (0.85,1.20) | 1.05 (0.88,1.26) |
| Czech Republic | | **2.15 (1.84,2.51)** | **1.63 (1.39,1.91)** | **1.64 (1.40,1.92)** |
| Hungary | | **0.73 (0.60,0.89)** | **0.59 (0.48,0.73)** | 0.84 (0.70,1.02) |
| Poland | | **1.23 (1.06,1.44)** | **0.78 (0.64,0.95)** | 0.92 (0.76,1.11) |
| Romania | | **0.49 (0.39,0.60)** | **0.76 (0.61,0.93)** | **0.74 (0.61,0.89)** |
| Slovakia | | 1.15 (0.96,1.38) | 1.15 (0.96,1.38) | 0.91 (0.76,1.09) |
| **Southern Europe** | | | | |
| Croatia | | **1.19 (1.01,1.40)** | **1.32 (1.09,1.58)** | 0.84 (0.71,1.01) |
| Cypress-TCC | | 0.91 (0.70,1.19) | | |
| Cyprus | | **1.67 (1.33,2.09)** | 0.87 (0.67,1.12) | 0.84 (0.66,1.08) |
| Greece | 1.00 (0.84,1.18) | **1.88 (1.62,2.19)** | 0.93 (0.78,1.11) | 1.14 (0.96,1.35) |
| Italy | 0.98 (0.84,1.16) | **0.45 (0.37,0.56)** | **0.46 (0.37,0.58)** | **0.62 (0.50,0.75)** |
| Malta | | **0.41 (0.29,0.59)** | 0.81 (0.58,1.13) | 1.05 (0.80,1.39) |
| Portugal | **0.59 (0.47,0.74)** | **0.40 (0.31,0.52)** | **0.49 (0.39,0.61)** | **0.69 (0.57,0.82)** |
| Slovenia | | **1.21 (1.03,1.42)** | **0.62 (0.51,0.77)** | 0.91 (0.76,1.08) |
| Spain | **0.71 (0.59,0.85)** | **0.58 (0.48,0.71)** | **0.46 (0.37,0.58)** | **0.62 (0.50,0.76)** |
| Turkey | | 0.92 (0.74,1.15) | | |
| **Gender** | | | | |
| Men (ref) | 1.00 | 1.00 | 1.00 | 1.00 |
| Women | **0.69 (0.61,0.77)** | **0.72 (0.65,0.8)** | **0.74 (0.67,0.83)** | **0.75 (0.68,0.84)** |
| **Age** | | | | |
| 18–24 years | 0.91 (0.72,1.15) | 0.97 (0.78,1.21) | **0.68 (0.53,0.87)** | 0.82 (0.63,1.06) |
| 25–34 years (ref) | 1.00 | 1.00 | 1.00 | 1.00 |
| 35–44 years | 1.00 (0.83,1.21) | 0.93 (0.79,1.09) | 0.93 (0.78,1.12) | 0.97 (0.81,1.16) |
| 45–54 years | 1.04 (0.85,1.26) | 1.06 (0.89,1.25) | 1.02 (0.85,1.22) | 0.97 (0.80,1.16) |
| 55–64 years | **0.79 (0.64,0.98)** | **0.73 (0.60,0.88)** | **0.81 (0.67,0.98)** | 0.94 (0.79,1.13) |
| 65 years and older | 0.97 (0.80,1.18) | 0.92 (0.76,1.11) | **0.74 (0.62,0.89)** | 0.85 (0.71,1.00) |
| **Marital Status** | | | | |

(*Continued*)

**Table 2.** (Continued)

| Category | Multivariate model OR (95%CI) of sitting >7.5 hours per day | | | |
|---|---|---|---|---|
| | 2002 | 2005 | 2013 | 2017 |
| Single (ref) | 1.00 | 1.00 | 1.00 | 1.00 |
| Married/De-facto | **0.73 (0.62,0.85)** | 0.90 (0.78,1.05) | 0.92 (0.79,1.06) | **0.77 (0.67,0.89)** |
| Separated/Divorced | 0.99 (0.79,1.25) | 1.16 (0.93,1.45) | 1.04 (0.83,1.30) | 0.82 (0.66,1.03) |
| Widowed | 1.23 (0.96,1.57) | **1.3 (1.03,1.66)** | **1.32 (1.06,1.65)** | **1.44 (1.17,1.78)** |
| **Age when stopped Education** | | | | |
| 15 years and less (ref) | 1.00 | 1.00 | 1.00 | 1.00 |
| 16 to 19 years | 1.02 (0.87,1.19) | **1.19 (1.03,1.38)** | 0.92 (0.78,1.08) | **0.80 (0.69,0.94)** |
| 20 years and over | **2.04 (1.73,2.40)** | **2.10 (1.80,2.45)** | **1.64 (1.39,1.93)** | **1.19 (1.02,1.39)** |
| Still Studying | **2.92 (2.24,3.80)** | **3.80 (2.91,4.95)** | **2.63 (1.97,3.53)** | **2.18 (1.63,2.93)** |
| **Type of Community** | | | | |
| Rural/Village | **0.65 (0.56,0.75)** | **0.67 (0.59,0.77)** | **0.65 (0.56,0.74)** | **0.77 (0.67,0.89)** |
| Small/Mid-sized town | **0.77 (0.67,0.89)** | **0.85 (0.75,0.96)** | **0.77 (0.68,0.88)** | **0.83 (0.74,0.95)** |
| Large Town (ref) | 1.00 | 1.00 | 1.00 | 1.00 |
| **Physical Activity Profile** | | | | |
| First Quartile—Low PA (ref) | 1.00 | 1.00 | 1.00 | 1.00 |
| Second Quartile | **0.61 (0.52,0.72)** | **0.69 (0.61,0.80)** | **0.77 (0.67,0.89)** | **0.64 (0.56,0.73)** |
| Third Quartile | **0.40 (0.35,0.47)** | **0.47 (0.41,0.54)** | **0.53 (0.46,0.61)** | **0.47 (0.40,0.54)** |
| Fourth Quartile—High PA | **0.23 (0.19,0.28)** | **0.24 (0.21,0.28)** | **0.35 (0.30,0.42)** | **0.23 (0.20,0.28)** |
| **Occupation*** | | | | |
| Self-employed | **2.1 (1.61,2.74)** | **2.18 (1.74,2.73)** | **2.22 (1.74,2.84)** | **2.21 (1.71,2.84)** |
| Manager | **4.6 (3.65,5.8)** | **4.33 (3.56,5.27)** | **4.61 (3.72,5.7)** | **4.37 (3.55,5.38)** |
| Other white collar | **5.8 (4.67,7.19)** | **5.94 (4.92,7.18)** | **6.92 (5.65,8.48)** | **5.41 (4.45,6.58)** |
| Manual worker (ref) | 1.00 | 1.00 | 1.00 | 1.00 |
| House person | 1.23 (0.95,1.60) | 1.24 (0.96,1.59) | 0.75 (0.52,1.07) | **1.64 (1.19,2.26)** |
| Unemployed | **1.79 (1.32,2.43)** | **1.58 (1.21,2.08)** | 1.30 (0.99,1.71) | **1.95 (1.49,2.57)** |
| Retired | **2.06 (1.66,2.56)** | **1.83 (1.45,2.30)** | **2.04 (1.64,2.54)** | **2.35 (1.90,2.91)** |
| Student | **4.92 (3.70,6.53)** | **5.68 (4.35,7.40)** | **4.92 (3.66,6.61)** | **5.26 (3.90,7.11)** |
| **Country (region)**** | | | | |
| Northern Europe | | **1.45 (1.24,1.69)** | **2.24 (1.90,2.65)** | **1.51 (1.29,1.76)** |
| Western Europe | | **1.54 (1.34,1.77)** | **2.26 (1.94,2.64)** | **1.62 (1.40,1.87)** |
| Eastern Europe | | **1.48 (1.29,1.70)** | **1.73 (1.48,2.02)** | **1.44 (1.25,1.67)** |
| Southern Europe (Ref) | | 1.00 | 1.00 | 1.00 |

*An additional model was conducted replacing education with occupation and adjusting for all other covariates as per the previous model. Only the results for occupation are presented for this model.

** An additional model was conducted replacing country with a country variable with clustering regions and adjusting for all other covariates as per the previous model. The 2002 results are not presented due to lower number of countries included in each region.

Notes: bold numbers represent a significant effect of p<0.05. Abbreviations: OR: odds ratio; ref: reference category: PA: physical activity

Eurobarometer data [15] was also limited by this change in methodology but has not shown further declines to 2017, which showed slightly higher sitting time estimates than the 2013 survey.

Furthermore, we identified stable correlates associated with high levels of self-reported sitting time. Respondents that consistently reported high levels of sitting time were more likely to be men, more educated, living in urban areas, white collar workers, retired people, students, and people with lower PA levels. This is similar to previous systematic reviews on correlates of

**Table 3. Multivariate odds ratio (OR) of sitting more than 7.5 hours per day <u>and</u> being in the least active physical activity quartile, by country and socio-demographic characteristics for each survey year.**

| Category | Multivariate model OR (95% CI) of sitting >7.5 hours per day and being in the least active quartile | | | |
|---|---|---|---|---|
| | 2002 | 2005 | 2013 | 2017 |
| **Country** (ref: all other countries) | | | | |
| **Northern Europe** | | | | |
| Denmark | **2.03 (1.48,2.77)** | **2.23 (1.62,3.08)** | **1.75 (1.32,2.33)** | **1.75 (1.29,2.37)** |
| Estonia | | 1.42 (0.97,2.06) | **1.60 (1.21,2.13)** | 1.3 (0.96,1.76) |
| Finland | 1.10 (0.83,1.46) | **1.32 (1.01,1.73)** | 1.06 (0.75,1.51) | **0.71 (0.51,0.99)** |
| Ireland | **0.57 (0.43,0.76)** | **0.57 (0.43,0.75)** | **0.61 (0.42,0.87)** | **0.66 (0.50,0.87)** |
| Latvia | | 1.02 (0.74,1.40) | 1.25 (0.90,1.73) | 1.27 (0.94,1.73) |
| Lithuania | | 0.8 (0.56,1.15) | 1.06 (0.80,1.40) | 0.87 (0.66,1.15) |
| Sweden | 1.20 (0.93,1.56) | 1.04 (0.81,1.33) | 1.15 (0.82,1.62) | 1.41 (0.99,2.00) |
| UK | 1.15 (0.89,1.50) | 1.20 (0.91,1.59) | **1.45 (1.12,1.88)** | 0.78 (0.59,1.04) |
| **Western Europe** | | | | |
| Austria | **0.73 (0.54,0.98)** | 0.76 (0.57,1.02) | 0.89 (0.65,1.20) | 0.80 (0.61,1.05) |
| Belgium | 1.08 (0.85,1.37) | 1.17 (0.92,1.50) | 1.06 (0.83,1.34) | 1.18 (0.91,1.54) |
| France | **0.71 (0.56,0.90)** | **0.63 (0.48,0.83)** | 1.02 (0.78,1.34) | 0.87 (0.66,1.14) |
| Germany | **1.45 (1.14,1.84)** | **1.81 (1.34,2.43)** | 1.23 (0.93,1.65) | 1.25 (0.93,1.67) |
| Luxembourg | 1.39 (0.95,2.03) | 1.34 (0.87,2.06) | 1.11 (0.74,1.68) | 1.24 (0.77,2.01) |
| Netherlands | 1.19 (0.78,1.82) | **3.45 (2.21,5.38)** | **3.23 (2.38,4.40)** | **2.95 (2.19,3.98)** |
| **Eastern Europe** | | | | |
| Bulgaria | | 0.92 (0.63,1.33) | **1.54 (1.19,1.99)** | 1.21 (0.92,1.58) |
| Czech Republic | | **3.15 (2.26,4.39)** | **1.59 (1.24,2.03)** | **1.9 (1.51,2.39)** |
| Hungary | | 0.95 (0.69,1.32) | **0.63 (0.47,0.85)** | 0.86 (0.67,1.11) |
| Poland | | **1.53 (1.13,2.06)** | **0.69 (0.53,0.90)** | 0.89 (0.69,1.16) |
| Romania | | **0.28 (0.18,0.44)** | 0.82 (0.60,1.13) | **0.44 (0.31,0.60)** |
| Slovakia | | 0.87 (0.60,1.27) | 1.06 (0.79,1.41) | 0.95 (0.73,1.22) |
| **Southern Europe** | | | | |
| Croatia | | **1.54 (1.14,2.08)** | 1.22 (0.90,1.65) | 1.01 (0.78,1.30) |
| Cypress-TCC | | 1.02 (0.71,1.47) | | |
| Cyprus | | **2.08 (1.50,2.88)** | 1.00 (0.74,1.37) | 0.93 (0.69,1.26) |
| Greece | 0.89 (0.67,1.18) | **1.88 (1.40,2.53)** | 1.06 (0.82,1.37) | **1.29 (1.02,1.62)** |
| Italy | 1.16 (0.90,1.48) | **0.39 (0.28,0.54)** | **0.37 (0.27,0.51)** | **0.66 (0.51,0.85)** |
| Malta | | **0.34 (0.21,0.54)** | 0.72 (0.47,1.11) | 1.12 (0.80,1.58) |
| Portugal | **0.53 (0.37,0.77)** | **0.29 (0.19,0.43)** | **0.54 (0.40,0.73)** | 0.81 (0.65,1.00) |
| Slovenia | | 1.32 (0.94,1.84) | **0.63 (0.46,0.85)** | 0.93 (0.71,1.23) |
| Spain | **0.66 (0.49,0.89)** | **0.57 (0.42,0.76)** | **0.43 (0.28,0.65)** | **0.53 (0.37,0.76)** |
| Turkey | | **0.65 (0.44,0.96)** | | |
| **Gender** | | | | |
| Men (ref) | 1.00 | 1.00 | 1.00 | 1.00 |
| Women | **0.65 (0.54,0.78)** | **0.61 (0.51,0.74)** | **0.73 (0.62,0.86)** | **0.82 (0.7,0.96)** |
| **Age** | | | | |
| 18–24 years | 0.85 (0.57,1.28) | 0.82 (0.54,1.26) | 0.68 (0.44,1.05) | 1.11 (0.69,1.78) |
| 25–34 years (ref) | 1.00 | 1.00 | 1.00 | 1.00 |
| 35–44 years | 0.83 (0.61,1.14) | 0.74 (0.54,1.00) | 0.88 (0.65,1.20) | 1.17 (0.87,1.59) |
| 45–54 years | 1.11 (0.80,1.53) | 1.16 (0.85,1.57) | 1.14 (0.84,1.54) | 1.09 (0.80,1.47) |
| 55–64 years | **0.69 (0.49,0.99)** | 0.76 (0.54,1.08) | 0.90 (0.66,1.23) | 1.22 (0.91,1.63) |
| 65 years and older | 1.02 (0.75,1.39) | 0.94 (0.68,1.31) | 0.98 (0.74,1.31) | 1.20 (0.91,1.57) |

*(Continued)*

**Table 3.** (Continued)

| Category | Multivariate model OR (95% CI) of sitting >7.5 hours per day and being in the least active quartile | | | |
| --- | --- | --- | --- | --- |
| | 2002 | 2005 | 2013 | 2017 |
| **Marital Status** | | | | |
| Single (ref) | 1.00 | 1.00 | 1.00 | 1.00 |
| Married/De-facto | **0.74 (0.57,0.95)** | 0.95 (0.72,1.25) | 0.91 (0.72,1.16) | **0.78 (0.61,0.98)** |
| Separated/Divorced | 1.32 (0.91,1.93) | 1.28 (0.85,1.94) | 1.01 (0.72,1.43) | 0.88 (0.62,1.26) |
| Widowed | 1.26 (0.86,1.83) | **1.55 (1.03,2.34)** | **1.44 (1.04,2.00)** | **1.42 (1.04,1.94)** |
| **Age when stopped Education** | | | | |
| 15 years and less (ref) | 1.00 | 1.00 | 1.00 | 1.00 |
| 16 to 19 years | 1.00 (0.79,1.26) | 1.25 (0.97,1.60) | 0.89 (0.71,1.11) | 0.83 (0.67,1.02) |
| 20 years and over | **1.83 (1.40,2.38)** | **2.03 (1.55,2.67)** | **1.44 (1.14,1.82)** | 1.08 (0.86,1.37) |
| Still Studying | **1.74 (1.07,2.83)** | **4.31 (2.51,7.41)** | **1.78 (1.05,3.00)** | 1.47 (0.85,2.54) |
| **Type of Community** | | | | |
| Rural/Village | **0.78 (0.62,0.98)** | 0.83 (0.66,1.05) | **0.59 (0.48,0.74)** | **0.77 (0.62,0.96)** |
| Small/Mid-sized town | 0.92 (0.73,1.16) | 0.92 (0.73,1.17) | **0.59 (0.48,0.72)** | **0.77 (0.63,0.94)** |
| Large Town (ref) | 1.00 | 1.00 | 1.00 | 1.00 |
| **Occupation**[*] | | | | |
| Self-employed | **1.92 (1.25,2.95)** | **2.28 (1.5,3.47)** | **2.53 (1.72,3.74)** | **2.14 (1.46,3.15)** |
| Manager | **2.98 (2.02,4.39)** | **4.84 (3.34,7.02)** | **3.98 (2.74,5.78)** | **3.07 (2.2,4.29)** |
| Other white collar | **4.28 (3.01,6.07)** | **7.26 (5.03,10.47)** | **5.62 (4.02,7.86)** | **4.63 (3.44,6.25)** |
| Manual worker (ref) | 1.00 | 1.00 | 1.00 | 1.00 |
| House person | 1.10 (0.73,1.65) | 1.49 (0.97,2.28) | 0.79 (0.48,1.31) | **1.99 (1.28,3.09)** |
| Unemployed | **1.77 (1.08,2.90)** | **1.88 (1.1,3.19)** | **1.60 (1.04,2.47)** | **2.10 (1.36,3.23)** |
| Retired | **1.97 (1.41,2.76)** | **2.22 (1.47,3.35)** | **2.33 (1.66,3.26)** | **2.36 (1.74,3.21)** |
| Student | **2.46 (1.45,4.17)** | **8.11 (4.6,14.29)** | **3.27 (1.89,5.66)** | **3.07 (1.76,5.37)** |
| **Country (region)** [**] | | | | |
| Northern Europe | | **2.11 (1.62,2.75)** | **2.82 (2.18,3.66)** | 1.27 (0.99,1.61) |
| Western Europe | | **1.89 (1.48,2.41)** | **2.57 (2.03,3.26)** | **1.61 (1.30,2.00)** |
| Eastern Europe | | **1.95 (1.53,2.50)** | **1.84 (1.46,2.31)** | **1.29 (1.05,1.58)** |
| Southern Europe (Ref) | | 1.00 | 1.00 | 1.00 |

[*]An additional model was conducted replacing education with occupation and adjusting for all other covariates as per the previous model. Only the results for occupation are presented for this model.

[**] An additional model was conducted replacing country with a country variable with clustering regions and adjusting for all other covariates as per the previous model. The 2002 results are not presented due to lower number of countries included in each region.

Notes: bold numbers represent a significant effect of p<0.05. Abbreviations: OR: odds ratio; ref: reference category

sedentary behavior [28–30]. It seems that southern Europe showed lower odds for being sedentary compared to all other European regions. The correlates that were consistently associated with high levels of sitting time and physical inactivity were generally the same as those found for high sitting levels, although not all were significantly associated in all four surveys.

Total sitting time seemed reasonably stable, although it is possible that domain-specific sitting is shifting. The results of a Danish workforce study between 1990–2010 suggested that occupational sitting gradually increased, especially in people with high socioeconomic status [31]. The national Dutch time use survey showed that occupational time increased between 1975–2005, although the study was unable to estimate occupation sitting. This increase in occupational time was accompanied with a decrease in non-occupational time as well as a decrease in non-occupational sitting time. However, non-occupational sitting time remained

relatively constant (~60% of all non-occupation time) over this 30-year period [32]. Although total time spent sitting has been stable over the past decade, there may have been a shift to more sitting at work and less sitting outside work. However, there is a lack of population data that allows the study of temporal changes in total and domain-specific sitting time.

The strongest associations were found for current occupation, with white collar workers and students being five times more likely to report high sitting. Additionally, we observed that higher educated people sit more, likely in more sedentary office work and study. This suggests that occupational sitting is a major contributor to total sitting time and pushes many people with a sitting job into the high sitting category. However, this group has reduced its "high-sitting/least-active" proportion over time, indicating possible early responsiveness to sitting reduction messages. Further, some advantaged workplaces have increased awareness of sedentary behavior, and employers have provided activity permissive desks, or behavioral coaching [33]. The other major contributor to total sitting time is leisure sitting time, which has been shown to be ~85% of leisure time in Dutch adults [32]. Given the high proportion of leisure time that is spent sitting, this is a prime target across the adult population for intervention studies aiming to replace sitting time with more movement. Especially, since screen-based sitting might have further increased sitting time over the past decade due to the rapid increase in availability of screen-based technologies

Adults categorized in the lowest PA quartile were four times more likely to report high sitting. This group comprises 7–10% of the European population, and is of particular importance since they have the markedly increased joint risk of sitting too much and being inactive [9]. Based on this study, similar groups that are at risk for high sitting are also at risk to be in the "high-sitting/least-active" group. This suggests an increased need to target these groups in prevention programs. Although, a recent review highlighted that combined PA and sitting time interventions are less effective in reducing sitting time than interventions that primarily focus on sitting time [34]. It remains important that future intervention developers realize that combined high sedentary time and physical inactivity might be considered as the same behaviors by the community [35], and interventions should distinguish between them, as substantial improvements in moderate to vigorous PA only result in small changes in sitting time. In addition, it might be helpful to develop clear thresholds such as those available for PA (e.g. 10,000 steps), to provide technological opportunities to self-monitor sedentary behavior accurately and to receive tips or advice on how people can reduce total volume of sitting and break up sitting time [34]. Although, specific public health guidelines around sedentary behavior require further development of the evidence base [36]. The effective integration of the "move more and sit less" message in intervention programs is a challenge that needs to be addressed in future interventions.

To date, the majority of sedentary behavior interventions have targeted high educated, white collar workers [37]. Based on the current study these are groups that are at increased risk of sitting too much and to some extent are also inactive. However, it is important not to overlook other high risk groups, such as men, unemployed people, retired people and students. For example, occupational health interventions might also focus on sitting jobs with a high proportion of low socio-economic status men (e.g., truck drivers, factory workers), for whom evidence is limited [38], as it is for unemployed and retired people [39].

## Strengths and limitations

A strength of the current study is the comparison of serial Eurobarometer surveys over a 15-year period across 28 EU member states, which allows us to identify the stability of correlates of "high sitting" and of "high-sitting/least-active" populations. However, these surveys have some limitations.

Firstly, in 2013 and 2017 the answering format of the IPAQ-short questionnaire was changed from an open-ended question to pre-specified categories. As very broad categories were used for PA, we were not able to calculate time spent in PA in relation to the public health guidelines, but used quartiles of reported days consistently across surveys instead. It is therefore possible we misclassified people, although with quartiles we believe we are at the conservative side given that more than one third of the European adult population is inactive [40]. Furthermore, it is possible that the change in answer categories influenced participant responses to the sitting item, especially as the highest category was set at >8.5 h/d. As a consequence, people might have underestimated their sitting time as respondents might have been less inclined to choose the highest response category (i.e. "I sit a lot but definitely not in the highest category"). Future Eurobarometer surveys could be improved by using the original validated IPAQ questions [22]. Future Eurobarometer surveys and other population-based studies are needed to draw definitive trend conclusions, and might additionally include device-based sitting measures as well as domain-specific measurements of sitting time to further understand sitting trends, for which example questionnaires are available at the sedentary behaviour research network (SBRN).

Secondly, the Eurobarometer claims to provide representative data for each country and that selection biases are relatively constant over the years. However, we observed variation in the results between years within individual countries (i.e. increases in age and years of education), which might have resulted from changes in data collection within countries.

Thirdly, the self-reported nature of the questionnaire allowed the collection of a large volume of data, but in comparison to device-based measurement of sedentary behavior and PA, self-reported data is prone to recall and social desirability biases [41, 42], although we do not have information if this bias was differential over time.

Fourthly, sitting time was assessed using a single item asking about a usual day, which prevented us to draw any conclusions on the different correlates for type of day (weekday or weekend) [28], and different domains, such as leisure time, transportation time, household time or occupational time.

Finally, the number of correlates in the Eurobarometer surveys are restricted to general personal characteristics and studies including contextual, policy and individual correlates would better identify factors associated with "high-sitting" and "high-sitting/least-active".

## Conclusion

Overall, sitting time remained relatively constant over the 15-year study period. Regular monitoring of sedentary behavior and PA is needed to monitor population trends and benchmark national and international policies, but should keep methods and measures identical over time. The present study reports stable observations that men, higher educated people, urban residents, white collar workers, unemployed people, retired people, students and those people with lower PA levels are more likely to report high levels of sitting time, but also high levels of sitting combined with low levels of PA. Addressing these factors not in isolation, but taking into account social, environmental and policy factors is most likely to result in significant changes in sitting time and PA [28].

## Supporting information

**S1 Table. Total population, high sitters and high sitters-least active by country for all four surveys.**
(DOCX)

## Acknowledgments

We would like to thank all respondents throughout the European Union who have participated in the Eurobarometer survey.

## Author Contributions

**Conceptualization:** Joanne Gale, Adrian Bauman, Hidde P. van der Ploeg.

**Formal analysis:** Joanne Gale.

**Supervision:** Hidde P. van der Ploeg.

**Writing – original draft:** Judith G. M. Jelsma.

**Writing – review & editing:** Judith G. M. Jelsma, Joanne Gale, Anne Loyen, Femke van Nassau, Adrian Bauman, Hidde P. van der Ploeg.

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
