## [Decision Letter · Decision Letter 0]

5 Sep 2019

PONE-D-19-17578

Time trends between 2002 and 2017 in correlates of self-reported sitting time in European adults

PLOS ONE

Dear Dr Jelsma,

Thank you for submitting your manuscript to PLOS ONE. After careful consideration, we feel that it has merit but does not fully meet PLOS ONE’s publication criteria as it currently stands. Therefore, we invite you to submit a revised version of the manuscript that addresses the points raised during the review process.

A couple of major comments were addressed. Taking them into account would improve the paper. Hoping it will help the authors for the revision.

We would appreciate receiving your revised manuscript by Oct 20 2019 11:59PM. To enhance the reproducibility of your results, we recommend that if applicable you deposit your laboratory protocols in protocols.io, where a protocol can be assigned its own identifier (DOI) such that it can be cited independently in the future. For instructions see: http://journals.plos.org/plosone/s/submission-guidelines#loc-laboratory-protocols

We look forward to receiving your revised manuscript.

Kind regards,

Anne Vuillemin

Academic Editor

PLOS ONE

Reviewers' comments:

Reviewer's Responses to Questions

**Comments to the Author**

1. Is the manuscript technically sound, and do the data support the conclusions?

Reviewer #1: Yes

2. Has the statistical analysis been performed appropriately and rigorously? 

Reviewer #1: Yes

3. Have the authors made all data underlying the findings in their manuscript fully available?

Reviewer #1: Yes

4. Is the manuscript presented in an intelligible fashion and written in standard English?

Reviewer #1: Yes

5. Review Comments to the Author

Reviewer #1: This is a clearly written manuscript examining an interesting research question: what are the time trends (4 measurement points over a 15-year period) in highly sitting and its correlates among adults from 28 European countries. There are some limitations, mainly related to the measures of the outcomes, but the authors address them fairly. I only have a couple of major comments.

- It would be informative to the reader to explicitly spell out why both outcomes are important or why you look at both outcomes and not only at ‘high sitting – least active’?

- How were time trends evaluated? Explain how ‘time’ is being used in the analyses?

- Provide some more explicit information to improve understandability (suggestions below), especially in the discussion.

Some suggestions are formulated here to further improve the manuscript and guide the reader better.

ABSTRACT:

- The authors may want to add a sentence regarding the background of this purpose. For example, briefly explain why these TWO outcomes (instead of looking at high sitting – low activity only) are important to be investigated.

- Add information on sample size and some participant characteristics if possible.

- Isn’t walking also measured in the IPAQ? Was it included to determine the PA quartiles? Clarify what is being used to determine ‘physical activity’. Does this include the days of walking, MPA and VPA, or only MPA and VPA?

- Explain what type of analyses were used to get the odds ratios and explain how ‘time’ is being used (e.g. were the analyses done separately for every year?).

- It might be useful to provide a range of the odds ratios in the results section.

- The authors might want to end the conclusion with a statement that is based on the current findings.

INTRODUCTION:

- Last sentence of paragraph 1: add what kind of risks are increased (related to ref 9).

- Maybe consider to briefly clarify the use of various terms “sedentary behaviour” and “sitting time”. Are they being used interchangeably here?

- It would be informative to the reader to explicitly spell out why both outcomes are important or why you look at both outcomes and not only at ‘high sitting – least active’?

- Last paragraph: consider to number the different aims and formulate some hypotheses.

METHODS:

- Please provide a reference for the statement that the reported number of days in PA items are a good proxy of total PA levels.

- Is there information available on income?

- Add the number of models that were being tested. Was this 8, two for every survey year?

- How were time trends evaluated? Explain how ‘time’ is being used in the analyses?

RESULTS:

- Sampling: which tests were used to examine for example the increase in higher educated respondents etc? Was this explained in the analyses section?

- Sampling: could you also say that there was an increase in the proportion of older respondents?

- Sampling: ‘…the mean number of total days OF PA IN THE LAST WEEK in the least active PA quartile…’ the authors might want to add the words in capital.

- Table 1: the authors might want to add the number of days to the PA quartile variables.

- Table 1: It could be informative to provide the different countries here as well? Is there a reason why this variable is not presented in table 1?

- Table 1: it would be informative to provide some more categories of the sitting time variable here. For example, is the majority of the sample sitting between 2-3 hrs, or 5-7.5 hrs daily?

- Table 2: Netherlands: there seems to be a big difference in odds ratio between 2002 and 2017. Is this correct? How could this be explained?

- Table 2: just a suggestion to place the ‘country (region)’ variable below the country variable at the top of the table.

- Prevalence of high sitting time: are these findings statistically being tested?

- Prevalence of high sitting time: add the odds ratio (between brackets) for all survey years. Or can these figures be presented in the table?

- Prevalence of high sitting time: ‘…we did not observe a difference in 2005’. Compared to when? It is not clear what survey year is being referred to.

- Correlates of “high sitting time”: It is not clear about what year the results are. Do you consider “consistently” as when the results are the same over the 4 survey years? Please explain a bit better to the reader.

- Correlates of “high sitting time”: suggestion to always add “compared to xxx” when reporting the results of the logistic regression analyses.

- Correlates of “high sitting time”: suggestion to guide the reader better by providing some odds ratios in the text as well.

- Correlates of “high sitting time”: was there a trend toward significance for widowed individuals to have higher odds compared to singles?

- Correlates of “high sitting time and least-active”: suggestion to always add “compared to xxx” when reporting the results of the logistic regression analyses.

DISCUSSION:

- First sentence: consider to add ‘HIGH LEVELS OF’ before ‘sitting time’. Idem for the first sentence of the second paragraph.

- First paragraph: even though this concerns the discussion section, it would be good to provide some figures to guide the reader better. Idem for the paragraph starting with ‘The strongest associations were found for current occupation…’

- Second paragraph: the authors might want to add some country results here as well. Or make comparisons with other continents, if possible?

- Can the authors elaborate more on screen-based sitting and recent technologies that might have changed over the last 15 years, affecting leisure time sitting? Do the authors have any hypotheses on how this might have affected the current findings?

- Referring to a Dutch study (ref 30): is there information available from other European countries as well? The Netherlands seems to be a bit atypical in terms of sitting time levels compared to the other countries.

- Section regarding ‘clear thresholds such as those available for PA’: do the authors suggest having public health guidelines for sitting time? Can the authors reflect on this and maybe address existing literature regarding this matter, for example Stamatakis et al, Br J Sports Med 2019?

- Last paragraph: please provide a reference for the first sentence (majority of SB interventions have targeted high educated, white-collar workers).

STRENGTHS AND LIMITATIONS:

- Please expand explicitly on the impactions of not being able to calculate time spent in PA in relation to public health guidelines. How could this have affected the current results?

- Idem for the sitting item, explicitly say what this means for the prevalence of high sitting.

- Do the authors have specific domain-specific measurements of sitting time in mind or would they recommend specific ones?

6. PLOS authors have the option to publish the peer review history of their article (what does this mean?). If published, this will include your full peer review and any attached files.

Reviewer #1: No

---

## [Author Response · Author response to Decision Letter 0]

15 Oct 2019

PONE-D-19-17578

Time trends between 2002 and 2017 in correlates of self-reported sitting time in European adults

PLOS ONE

Dear Dr. Vuillemin,

Thank you for the opportunity to revise the manuscript entitled “Time trends between 2002 and 2017 in correlates of self-reported sitting time in European adults”. We are grateful to you and to the reviewer for the time involved in reading our manuscript and for the helpful comments and suggestions to help us improve the manuscript. Below we address the comments of the reviewer in a point by point response.

Response to reviewer #1

1. It would be informative to the reader to explicitly spell out why both outcomes are important or why you look at both outcomes and not only at ‘high sitting – least active’?

Response: We examined ‘high sitting’ because we know that ‘high sitting’ is a risk for negative health outcomes, even in people who engage in high levels of physical activity. We examined ‘high sitting – least active’, because this combination of behaviors is associated with the highest risk of negative health outcomes. As both outcomes can identify target populations for public health interventions, we have included both. We tried to explain this more clearly:

Line 43-47: High volumes of sedentary behavior are associated with an increased risk of all-cause mortality [2-9], cardiovascular mortality [3-5, 8, 9], cardiovascular disease [3, 5, 8] and type 2 diabetes [3-6, 8]. Even though, physical activity attenuates the risk of high volumes of sedentary behavior, very high levels of physical activity are needed to fully compensate the risks of sedentary behavior [9].

Line 64-66: We examined factors that were consistent correlates of high sitting time, as well as the combination of high sitting time and low levels of PA, over all four time points. Both groups have merit for future public health interventions. 

2. - How were time trends evaluated? Explain how ‘time’ is being used in the analyses?

Response: We evaluated time trends in multivariate models, considering ‘time’ as a factor in the model. A time by correlate interaction term was included (country or other correlate) to calculate the odds ratios at each year by correlate combination. To clarify we added the following lines: 

Line 125-127: To evaluate time trends ‘time’ was considered a factor in the model. A time by correlate interaction term was included (country or other correlate) to calculate the odds ratios at each year by correlate combination. 

3. - Provide some more explicit information to improve understandability (suggestions below), especially in the discussion.

ABSTRACT:

3a. - The authors may want to add a sentence regarding the background of this purpose. For example, briefly explain why these TWO outcomes (instead of looking at high sitting – low activity only) are important to be investigated.

Response: We added the following sentence to Line 17-20: Background: This study explores trends in the prevalence of high sitting time and its correlates among “high sitting” and “high sitting-least active” European adults from 2002 to 2017. Both groups have merit for future public health interventions to prevent development of a range of prevalent non-communicable diseases.

3b. - Add information on sample size and some participant characteristics if possible.

Response: We added the following sentence to Line 21-23: : Data collected in the 2002 (15 countries), 2005 (30 countries), 2013 (28 countries) and 2017 (28 countries) Eurobarometer surveys were used, including around 15,000 respondents in 2002 and >26,000 respondents in the other years.

3c. - Isn’t walking also measured in the IPAQ? Was it included to determine the PA quartiles? Clarify what is being used to determine ‘physical activity’. Does this include the days of walking, MPA and VPA, or only MPA and VPA?

Response: We included days of walking, MPA and VPA, which is now reflected in the manuscript:

Line 24-26: Respondents in the lowest quartile of total reported days of physical activity (i.e. days walking, days in moderate activity, and days in vigorous activity) were defined as least active.

3d. - Explain what type of analyses were used to get the odds ratios and explain how ‘time’ is being used (e.g. were the analyses done separately for every year?).

Response: See response to comment 2. We added the following line to the abstract:

Line 27-28: Multivariate odds ratios of high sitting, and high sitting-least active were assessed by country and socio-demographic characteristics for each survey year using binary logistic regression analyses.

3e. - It might be useful to provide a range of the odds ratios in the results section.

Response: In light of the wordlimit and readability we have chosen not to include odds ratios in the abstract. 

3f. - The authors might want to end the conclusion with a statement that is based on the current findings.

Response: We added the following to the concluding sentence:

Line 36-37: ‘This study identified population sub-groups that need special attention in public health interventions to lower total sitting time.”

INTRODUCTION:

3g. - Last sentence of paragraph 1: add what kind of risks are increased (related to ref 9).

Response: We added the following addition to the first sentence of the introduction paragraph.

Line 47-49: Importantly, people who are also physically inactive (i.e. not meeting the WHO recommendations of at least 150 minutes of moderate-to-vigorous activity per week) are especially at risk for all-cause and cardiovascular disease mortality [9].

3h. - Maybe consider to briefly clarify the use of various terms “sedentary behaviour” and “sitting time”. Are they being used interchangeably here?

Response: the term ‘sedentary behavior’ is used in general, the term ‘sitting time’ is used related to the questions asked in research projects, in which sitting time is used as a proxy for sedentary behaviour. We added the following for clarification:

Line 56-57: In 2002, 2005, 2013 and 2017 questions about sitting time (i.e. proxy for sedentary behavior) were included.

3i. - It would be informative to the reader to explicitly spell out why both outcomes are important or why you look at both outcomes and not only at ‘high sitting – least active’?

Response: See response to comment 1.

3j. - Last paragraph: consider to number the different aims and formulate some hypotheses.

Response: We added the words ‘first’ and ‘secondly’ to the text to clarify both aims.

Line 63-66: “With the addition of the 2017 Eurobarometer we first studied 15-year trends in self-reported sitting time in European adults over four surveys. And secondly, we examined factors that were consistent correlates of high sitting time, as well as the combination of high sitting time and low levels of PA, over all four time points.”

We have not formulated hypotheses in advance.

METHODS:

3k. - Please provide a reference for the statement that the reported number of days in PA items are a good proxy of total PA levels.

Response: We have provided two references that show that days are a good proxy of total PA levels.

3l. - Is there information available on income?

Response: Unfortunately, income is not recorded in the Eurobarometer.

3m.- Add the number of models that were being tested. Was this 8, two for every survey year?

Response: We tested three models and recoded the time variable to calculate the odds ratios for each time point. See also response to comment 2.

Line 125-128: Due to collinearity between the education and occupation variables (i.e. both include the group ‘student’) two multivariate models were constructed, whereby only the results from the occupation variable are presented for the second multivariate model.

We added the following lines: 

Line 130-131: Furthermore, we conducted a third multivariate model replacing the country variable with a variable in which the countries were clustered in four geographical regions.

3n. - How were time trends evaluated? Explain how ‘time’ is being used in the analyses?

Response: See response to comment 2.

RESULTS:

3o.- Sampling: which tests were used to examine for example the increase in higher educated respondents etc? Was this explained in the analyses section?

Response: We did not test for a significant increase in education, although based on the results we see there is an increase from 21.4% in 2002 to 32.1% in 2017 for those who finished education at 20 years or over. This might be reflective of the increase in education level across the population over time or a sampling selection bias. We added the following line:

Line 290-294: However, we observed variation in the results between years within individual countries (i.e. increases in age and years of education), which might have resulted from changes in data collection within countries. 

3p. - Sampling: could you also say that there was an increase in the proportion of older respondents?

Response: Yes, over the years the mean age of the population increased. We commented on this:

Line 148-149: The mean age (SD) of the population was 46.1 (17.4), 47.7 (17.5), 50.1 (17.5) and 52.0 (17.7) in 2002, 2005, 2013 and 2017, respectively

This might be a result of an increase of the age of the population or a sampling selection bias. Similar to comment 3o, we added the following line: 

Line 292-294: However, we observed variation in the results between years within individual countries (i.e. increases in age and years of education), which might have resulted from changes in data collection within countries.

3q. - Sampling: ‘…the mean number of total days OF PA IN THE LAST WEEK in the least active PA quartile…’ the authors might want to add the words in capital.

Response: We added this as suggested.

3r. - Table 1: the authors might want to add the number of days to the PA quartile variables.

Response: We added the following lines to the method section, providing more information on the quartile distribution of days.

Line 98-100: We created quartiles of the sum of days (possible range 0-21 days) respondents reported doing PA across the three PA items (i.e. 1st quartile 0-5 days (low PA); 2nd quartile 6-7 days; 3rd quartile 8-12 days; 4th quartile >12 days (high PA)).

3s. - Table 1: It could be informative to provide the different countries here as well? Is there a reason why this variable is not presented in table 1?

Response: We think the table would be too overwhelming and reported the results of the different countries in appendix 1. 

3t. - Table 1: it would be informative to provide some more categories of the sitting time variable here. For example, is the majority of the sample sitting between 2-3 hrs, or 5-7.5 hrs daily?

Response: We know based on the literature that sitting more than 7.5 hours is a risk for negative health outcomes. As such, the distinction between sitting 2-3hrs or 5-7.5hrs is less interesting and less evidence based , and therefore we only focused on the high risk population in the current manuscript.

3u. - Table 2: Netherlands: there seems to be a big difference in odds ratio between 2002 and 2017. Is this correct? How could this be explained?

Response: The difference between 2002 and the other years could largely be explained by the sample of countries included. In 2002, the Netherlands is compared against 14 other countries, whereas in 2005, 2013 and 2017 the comparison is against 28 other countries. 

3v. - Table 2: just a suggestion to place the ‘country (region)’ variable below the country variable at the top of the table.

Response: Since the country region variable is a separate model we placed this in the lower part of the table as to distinguish between models and not to confuse the reader. 

3w. - Prevalence of high sitting time: are these findings statistically being tested?

Response: Yes. The presented odds ratios are the results of the statistical test. Survey year 2017 is the reference.

3x - Prevalence of high sitting time: add the odds ratio (between brackets) for all survey years. Or can these figures be presented in the table?

Response: The odds ratios are compared to 2017 (reference category). The odds ratios are presented for each survey year. 

3y. - Prevalence of high sitting time: ‘…we did not observe a difference in 2005’. Compared to when? It is not clear what survey year is being referred to.

Response: The survey is compared to 2017. We added the following line for clarity: 

Line 180-182: Over time we observe higher odds of “high-sitters/least-active” in 2002 (OR=1.27 (95%CI=1.13-1.43) compared to 2017, but we did not observe a difference in 2005 (OR=1.00 (95%CI=0.88-1.13) compared to 2017.

3z. - Correlates of “high sitting time”: It is not clear about what year the results are. Do you consider “consistently” as when the results are the same over the 4 survey years? Please explain a bit better to the reader.

Response: Yes the reviewer is correct. We consider consistently if the results are the same over the available surveys. We added the following lines for clarity: 

Line 131-133: In the result section we consider results ‘consistent’ if on all available time points they show statistically significant associations in the same direction.

3aa- Correlates of “high sitting time”: suggestion to always add “compared to xxx” when reporting the results of the logistic regression analyses.

Response: We have done this for each variable.

3ab- Correlates of “high sitting time”: suggestion to guide the reader better by providing some odds ratios in the text as well.

Response: For readability we decided against mentioning odds ratios in the text here, and instead refer to the tables.

3ac. - Correlates of “high sitting time”: was there a trend toward significance for widowed individuals to have higher odds compared to singles?

Response: There seems to be a trend in the later years. Although, we decided to only report consistent results (all available time points showed statistically significant associations in the same direction) and as such decided to report marital status shown to be a less clear correlate across the survey years.

3ad. - Correlates of “high sitting time and least-active”: suggestion to always add “compared to xxx” when reporting the results of the logistic regression analyses.

Response: We mention compared to (reference category) for each variable. For example, we added in Line 205-206: Women are less likely to demonstrate “high-sitters/least-active” compared to men.

DISCUSSION:

3ae. - First sentence: consider to add ‘HIGH LEVELS OF’ before ‘sitting time’. Idem for the first sentence of the second paragraph.

Response: We have changed this as suggested:

Line 210-212: In this study we explored the trends and correlates of high levels of self-reported sitting time and the combination of high levels of sitting time and physical inactivity in European adults assessed by the adults assessed by the Eurobarometer surveys across 28 EU Member States over four time points between 2002 and 2017.

Line 218: Furthermore, we identified stable correlates associated with high levels of self-reported sitting time.

3af.- First paragraph: even though this concerns the discussion section, it would be good to provide some figures to guide the reader better. Idem for the paragraph starting with ‘The strongest associations were found for current occupation…’

Response: We believe it would lower readability to present odds ratios in the discussion section. 

3ag. - Second paragraph: the authors might want to add some country results here as well. Or make comparisons with other continents, if possible?

Response: We added the following lines: 

Line 220-221: It seems that southern Europe showed lower odds for being sedentary compared to all other European regions.

3ah. - Can the authors elaborate more on screen-based sitting and recent technologies that might have changed over the last 15 years, affecting leisure time sitting? Do the authors have any hypotheses on how this might have affected the current findings?

Response: We added the following lines:

Line 246-248: The other major contributor to total sitting time is leisure sitting time, which has been shown to be ~85% of leisure time in Dutch adults [30]. Given the high proportion of leisure time that is spent sitting, this is a prime target across the adult population for intervention studies aiming to replace sitting time with more movement. Especially, since screen-based sitting might have further increased sitting time over the past decade due to the rapid increase in availability of screen-based technologies.

3ai.- Referring to a Dutch study (ref 30): is there information available from other European countries as well? The Netherlands seems to be a bit atypical in terms of sitting time levels compared to the other countries.

Response: Currently available studies measure sitting time mostly with a single question making it impossible to distinguish domain specific sitting time. We make mention of this in Line 287-291. The Dutch study we referred to looked into leisure time sitting in relation to total sitting time. Therefore, even though the Dutch are more sedentary compared to other countries leisure time sitting was reported as percentage and as such might be more reflective. 

3aj. - Section regarding ‘clear thresholds such as those available for PA’: do the authors suggest having public health guidelines for sitting time? Can the authors reflect on this and maybe address existing literature regarding this matter, for example Stamatakis et al, Br J Sports Med 2019?

Response: We thank the reviewer for this suggestion. We have added the following line and reference of Stamatakis et al. 2019.

Line 258-263: . In addition, it might be helpful to develop clear thresholds such as those available for PA (e.g. 10,000 steps), to provide technological opportunities to self-monitor sedentary behavior accurately and to receive tips or advice on how people can reduce total volume of sitting and break up sitting time [32]. Although, specific public health guidelines around sedentary behavior require further development of the evidence base [34]

3ak. - Last paragraph: please provide a reference for the first sentence (majority of SB interventions have targeted high educated, white-collar workers).

Response: We provide a reference for this statement (Peachey 2018).

STRENGTHS AND LIMITATIONS:

3al. - Please expand explicitly on the impactions of not being able to calculate time spent in PA in relation to public health guidelines. How could this have affected the current results?

Response: We added the following line: 

Line 278-282: As very broad categories were used for PA, we were not able to calculate time spent in PA in relation to the public health guidelines, but used quartiles of reported days consistently across surveys instead. It is therefore possible we misclassified people, although with quartiles we believe we are at the conservative side given that more than one third of the European adult population is inactive [38].

3am. - Idem for the sitting item, explicitly say what this means for the prevalence of high sitting.

Response: This might explicitly mean we might underestimate sitting time. We added the following line:

Line 284-286: As a consequence, people might have underestimated their sitting time as respondents might have been less inclined to choose the highest response category (i.e. “I sit a lot but definitely not in the highest category”).

3an. - Do the authors have specific domain-specific measurements of sitting time in mind or would they recommend specific ones?

Response: We added the following lines:

Line 287-290: Future Eurobarometer surveys and other population-based studies are needed to draw definitive trend conclusions, and might additionally include device-based sitting measures as well as domain-specific measurements of sitting time to further understand sitting trends, for which example questionnaires are available at the sedentary behaviour research network (SBRN).

---

## [Decision Letter · Decision Letter 1]

31 Oct 2019

Time trends between 2002 and 2017 in correlates of self-reported sitting time in European adults

PONE-D-19-17578R1

Dear Dr. Jelsma,

We are pleased to inform you that your manuscript has been judged scientifically suitable for publication and will be formally accepted for publication once it complies with all outstanding technical requirements.

With kind regards,

Anne Vuillemin

Academic Editor

PLOS ONE

---

## [Editor Report · Acceptance letter]

5 Nov 2019

PONE-D-19-17578R1 

Time trends between 2002 and 2017 in correlates of self-reported sitting time in European adults 

Dear Dr. Jelsma:

I am pleased to inform you that your manuscript has been deemed suitable for publication in PLOS ONE. Congratulations! Your manuscript is now with our production department. 

With kind regards,

on behalf of

Dr. Anne Vuillemin 

Academic Editor

PLOS ONE